# Biopsychosocial Factors for Chronicity in Individuals with Non-Specific Low Back Pain: An Umbrella Review

**DOI:** 10.3390/ijerph191610145

**Published:** 2022-08-16

**Authors:** Emilia Otero-Ketterer, Cecilia Peñacoba-Puente, Carina Ferreira Pinheiro-Araujo, Juan Antonio Valera-Calero, Ricardo Ortega-Santiago

**Affiliations:** 1Escuela Internacional de Doctorado, Universidad Rey Juan Carlos, 28922 Alcorcón, Spain; 2Physiotherapy Department, Mutua Universal Mugenat, 28001 Alcalá de Henares, Spain; 3Department of Psychology, Universidad Rey Juan Carlos, 28922 Alcorcón, Spain; 4Motor Science Institute, Federal University of Alfenas, Alfenas 37130-000, Brazil; 5Valtradofi Research Group, Department of Physiotherapy, Faculty of Health, Universidad Camilo José Cela, Villanueva de la Cañada, 28692 Madrid, Spain; 6Department of Physical Therapy, Occupational Therapy, Physical Medicine and Rehabilitation, Universidad Rey Juan Carlos, 28922 Alcorcón, Spain; 7Cátedra Institucional en Docencia, Clínica e Investigación en Fisioterapia: Terapia Manual, Punción Seca y Ejercicio Terapéutico, Universidad Rey Juan Carlos, 28922 Alcorcón, Spain

**Keywords:** chronic pain, prognosis, humans, low back pain, pain, risk factors, umbrella review

## Abstract

Low back pain (LBP) is a global and disabling problem. A considerable number of systematic reviews published over the past decade have reported a range of factors that increase the risk of chronicity due to LBP. This study summarizes up-to-date and high-level research evidence on the biopsychosocial prognostic factors of outcomes in adults with non-specific low back pain at follow-up. An umbrella review was carried out. PubMed, the Cochrane Database of Systematic Reviews, Web of Science, PsycINFO, CINAHL Plus and PEDro were searched for studies published between 1 January 2008 and 20 March 2020. Two reviewers independently screened abstracts and full texts, extracted data and assessed review quality. Fifteen systematic reviews met the eligibility criteria; all were deemed reliable according to our criteria. There were five prognostic factors with consistent evidence of association with poor acute–subacute LBP outcomes in the long term (high levels of pain intensity and disability, high emotional distress, negative recovery expectations and high physical demands at work), as well as one factor with consistent evidence of no association (low education levels). For mixed-duration LBP, there was one predictor consistently associated with poor outcomes in the long term (high pain catastrophism). We observed insufficient evidence to synthesize social factors as well as to fully assess predictors in the chronic phase of LBP. This study provides consistent evidence of the predictive value of biological and psychological factors for LBP outcomes in the long term. The identified prognostic factors should be considered for inclusion into low back pain explanatory models.

## 1. Introduction

Low back pain (LBP) is a common health condition with important implications for individuals, public health systems and economies [1]. It has been increasing worldwide since 1990 with the rise and aging of the population, with a higher prevalence among people between the ages of 40 and 80 [2,3]. In 2017, low back pain was the leading cause of years of disability, with over 570 million people affected at any one time [3], and it is likely to increase in low-income and middle-income countries in the next few decades [4]. Low back pain generates an impact on the quality of life of individuals [5,6] and on the economy, with direct healthcare costs [7] comparable to those of cardiovascular disease, cancer or mental health [8], as well as indirect costs related to the potential loss of work status [4,9].

Most people who experience LBP have non-specific low back pain (NSLBP), a heterogeneous presentation with variable prognosis, defined as low back pain not attributable to a recognizable and known specific pathology (e.g., infection, tumor, osteoporosis, lumbar spine fracture, structural deformity, inflammatory disorder, radicular syndrome or cauda equina syndrome) [2]. Currently, NSLBP is understood as a neurobiological and behavioral response to individual threat perception, rather than a disease [1]. The biopsychosocial model was embraced in 1977 [10], providing a framework to explain the complexity of disabling LBP and its multidimensional clinical reasoning up to the present day, incorporating the interaction between the social, psychological and biological dimensions of pain [11], context and behavioral conditioning [12]. 

Prognostic factors inform us about the likely course or outcome of a health condition over time and, thus, guide health professionals in decision-making and patient health education [13], in preventing the development and maintenance of chronic pain [14]. Since the publication of the last overview of systematic reviews on prognostic factors in individuals with LBP in 2009 [15], a considerable number of primary studies and systematic reviews on LBP predictors have been published, and, in turn, there has been substantial progress in search methods. However, most of these available systematic reviews have either focused on the analysis of a single prognostic factor or have done so regarding a specific outcome domain.

Therefore, the objective of this umbrella review is to display an up-to-date overview of high-level research evidence providing longitudinal data on biopsychosocial prognostic factors of outcomes in individuals with non-specific low back pain. 

## 2. Materials and Methods

One reviewer (EO) screened the titles identified by removing ineligible studies and, subsequently, two reviewers (EO and CP) independently examined all abstracts and full texts. Two other reviewers (CF and JV) extracted information using a standardized data extraction form and assessed the reliability of the reviews. Disagreements were discussed until consensus, and if consensus was not reached, a third reviewer (RO) was available.

### 2.1. Protocol and Registration

We followed the Umbrella Review Methodology Working Group [16] and considered the Preferred Reporting Items for Systematic Reviews and Meta-Analyses (PRISMA) recommendations [17] (Appendix A: PRISMA checklist). The protocol of the study was registered on PROSPERO 2020: CRD42020155081.

### 2.2. Criteria for Considering Reviews for the Overview

#### 2.2.1. Literature Search 

PubMed, the Cochrane Database of Systematic Reviews, Web of Science, PsycINFO, CINAHL Plus and PEDro were searched electronically for studies published between 1 January 2008 and 20 March 2020. Our search was limited to 2008 onwards, given that the previous “review of reviews” on LBP prognosis conducted a literature search until 2007 [15]. No restrictions were applied regarding the follow-up duration or language. The search strategy included low back pain (Cochrane Back and Neck Group recommended strategy) [18] and prognostic study method terms [19] (Appendix A: Search strategy for PubMed). The electronic search was implemented in several grey literature databases (NHS Evidence, Explore the British Library, Open Dissertations, TESEO, OpenGrey, CNCS-ISCiii, JBI COnNECT+, New York Academy of Medicine, New York, NY, USA). In addition, manual searches were also performed by tracking citations from the reference lists of all included reviews and relevant reviews in musculoskeletal (MSK) pain, as well as by contacting authors of included reviews.

#### 2.2.2. Review Selection

We selected systematic reviews, with or without meta-analysis, summarizing longitudinal observational studies that involved adult participants (≥18 years) at any point in the course of LBP (acute, subacute or chronic) or with mixed pain (i.e., other conditions such as neck or thoracic pain), only if most of the population (≥75%) underwent NSLBP or subgroup data were available for this condition, with baseline measures of at least one biological, psychological or social factor, as well as one predicating the primary outcomes (pain intensity, functional status, work participation and recovery) and, additionally, secondary outcomes (health-related quality of life, emotional distress, satisfaction with treatment and healthcare utilization); we included only those written in English or Spanish.

We excluded reviews involving a majority of individuals with LBP caused by specific pathologies or conditions (such as surgery or pregnancy); those assessing factors as mediators, moderators or their impact on treatment; those reporting only secondary outcomes; those based on a cross-sectional design; and narrative or methodological reviews.

#### 2.2.3. Data Extraction and Management

We recorded complete information about citations, populations, methods, prognostic factors and outcomes assessed. The results of the reviews were extracted separately for each duration of LBP symptoms: acute–subacute (≤3 months), chronic (>3 months) and mixed duration [20]. 

Likewise, results data were extracted for each primary outcome, according to the International Classification of Functioning, Disability and Health (ICF) framework [21]—pain intensity, functional status and work participation and recovery—which were considered together to synthesize the evidence of our outcomes of LBP results at follow-up. For the interpretation of the best available evidence, the secondary outcomes of health-related quality of life, emotional distress, satisfaction with treatment as well as healthcare utilization were collected and considered narratively. We categorized the results according to the follow-up time period—short-term (<3 months) and long-term (≥3 months)—along with the evidence that most improvements in pain, activity limitation and return to work occur within 3 months and thereafter recovery is lesser [20]. Moreover, since an unadjusted finding does not control for confounding factors, unlike the adjusted finding, we extracted all adjusted data, apart from the unadjusted data for a separately planned analysis, when possible [22]. 

When a systematic review presented data from several primary studies for the same factor, we reported the range (i.e., the lowest and highest value reported). In the event that the review described a meta-analysis, we presented the pooled estimate. Where several measurement instruments were reported for the same outcome, we selected the measure with greater evidence of validity and reliability for synthesis. Likewise, all dichotomous measures with more than one cut-off point were extracted, but the one showing the most significant association was used. In addition, the overlap of primary studies among the included reviews was recorded using citation matrices and excluded from our synthesis. The degree of overlapping studies was calculated using the Corrected Covered Area (CCA) method [23].

### 2.3. Methodological Quality Assessment of Included Reviews

We used the criteria developed by the SUPPORT and SURE collaborations, reported in a recent review published in The Cochrane Library [24]. It rates 14 criteria grouped under Section A—Identification, selection and critical appraisal of studies; Section B—Analysis; and Section C—Overall. Each item can be rated as follows: +, yes; ?, can’t tell/partially;—, no; NA, not applicable (e.g., no studies or data). In the last item, and considering the prior assessments of the criteria, the review is categorized as having (1) only minor limitations; (2) limitations that are important enough that it would be worthwhile to search for another systematic review and to interpret the results of this review cautiously, if no better review is available; (3) limitations that are important enough to compromise the reliability of the findings of the review and to prompt the exclusion of the review.

### 2.4. Data Synthesis

The characteristics of the included reviews were summarized descriptively. We also conducted a descriptive quantitative analysis (summary measure with a precision estimate) for each systematic review, according to the duration of LBP symptoms (acute, subacute, chronic and mixed duration) as well as length of follow-up (short and long term). 

To adequately compare findings across the reviews, we used odds ratio (and beta coefficients) statistics for synthesis. Results of a systematic review were considered consistent if ≥75% of the primary studies reporting on a factor rated the same direction of association with the outcome [25]. Thus, a factor was judged consistently associated with low back pain outcomes when it demonstrated a uniform association in the same direction by at least two reliable reviews, or at least half of them, and not contradicted by any other review [15]. The strength of association with outcomes was deemed weak (OR 1.01–1.49), moderate (OR 1.50–1.99) or strong (OR ≥ 2.0) [26], with moderate and strong strengths considered clinically relevant. 

Thus, a qualitative synthesis was performed given that the main purpose of this study was to present a summary of the current body of evidence based on systematic reviews of biopsychosocial prognostic factors in patients with LBP and also considering the heterogeneity of the data collected. In this way, we have described and discussed the extent of the main differences found in the results reported by the included reviews, as well as the aspects considered as probable explanatory factors for such heterogeneity, without performing additional subgroup or sensitivity analyses.

## 3. Results

### 3.1. Review Selection

A total of 2.721 citations were identified: 1.846 through electronic databases, 744 from grey literature databases and 131 from tracking citations and contact with authors. We evaluated 72 full-text publications, and 15 systematic reviews were eligible (see Figure 1). References from excluded full-text citations (*n* = 57) are reported in Appendix A. The conflicts of interest of the review authors are displayed in Appendix A.

Disagreements were resolved by consensus among reviewers twice during the selection process, four times during data extraction and twice during quality assessment, with non-intervention of the third reviewer.

### 3.2. Review Characteristics

Fifteen systematic reviews (257,208 participants) reported data on biopsychosocial prognostic factors and low back pain outcomes at follow-up, with four being general reviews [27,28,29,30] and 11 reviews focused on a single prognostic factor [31,32,33,34,35,36,37,38,39,40,41] (Table 1).

The reviews included studies performed in North America (15 reviews), South America (1 review), Europe (15 reviews), Oceania (11 reviews) and Asia (4 reviews). Most of the populations contained in the reviews displayed acute and subacute (57%, 146 studies from 10 reviews) and mixed-duration low back pain (39%, 100 studies from 7 reviews) from a clinical (61%, 156 studies) and occupational setting (36%, 93 studies). 

The publication date of the included reviews ranged from 2008 to 2019 and that of the included primary studies varied from 1981 to 2017 (Appendix A: Primary studies referenced in tables). Twelve reviews (80%) were published over 5 years ago (before 2015). The sample size in the reviews ranged from 219 [41] to 112.797 [29], with a mean of 17.147 (interquartile range (IQR): 3.535 to 11.330).

Regarding our outcome of LBP results, all primary outcomes were widely assessed across the 15 included reviews: work participation (60%, 9 reviews), functional status (60%, 9 reviews), pain intensity (60%, 9 reviews) and aspects of recovery (53%, 8 reviews). For secondary outcomes, only satisfaction with treatment and healthcare utilization results were reported by two primary studies, using *p*-values in two reviews [37,39], with insufficient evidence for interpretation. Only 5% of the primary studies included in the systematic reviews reported results within 3 months of follow-up (short term).

The main reasons for the review authors not pooling the results were the heterogeneity of the population, measures of prognostic factors, outcomes assessed and outcome measures, as well as the variety of statistical analyses. The most common estimates used to report the results across the reviews were odds ratios (OR), but beta coefficients (β), risk ratios (RR), prevalence ratios (RP), hazard ratios (HR), likelihood ratios (LR+/LR-) and *p*-values were also reported. Data from RR/RP, HR, LR+/LR- and *p*-values are provided in Appendix A.

### 3.3. Methodological Quality Assessment of Included Reviews

The results of our appraisal of the methodological quality (reliability) of the included reviews are shown in Appendix A. We judged all 15 included reviews to have only minor limitations. In general, there were few failures with regard to the selection criteria and critical appraisal of the risk of bias of the primary studies from the systematic reviews, with thirteen reviews partially meeting the comprehensive search strategy criterion. Likewise, there were few flaws regarding the analysis of the results, with three reviews showing limitations in the reporting of the characteristics and results and one review explaining the differences in the results.

### 3.4. Synthesis of Results 

Overall, forty-nine factors were reported across the included systematic reviews. The degree of overlap of the primary studies through the reviews was slight (CCA = 2.6%). 

#### 3.4.1. Acute–Subacute Phase of LBP

For most factors collected, there was insufficient evidence to synthesize the unadjusted data, so our summary of results was mainly based on adjusted results. Besides this, there were three factors (gender, previous history of LBP and pain radiating to the leg) for which there was also insufficient evidence to perform synthesis from adjusted data. In order to include as much evidence as possible [22], we combined both types of results (adjusted and unadjusted) to analyze the consistency of these variables.

Thus, there were 10 prognostic factors of outcomes at long-term follow-up provided by two or more systematic reviews with sufficiently similar data for comparability (OR/Beta), derived from seven systematic reviews (Table 2 and Table 3) [27,29,30,32,33,35,38]. Of these, five prognostic factors showed consistent evidence supporting their ability to predict poor long-term outcomes: high levels of pain intensity and disability, high emotional distress, negative recovery expectations and high work physical demands (Table 4). Another factor showed consistent evidence of no association with poor outcomes: low education levels. Each of these variables showed strengths of association ranging from weak to strong and outcomes reflecting clinical relevance (OR ≥ 1.50). Moreover, four factors demonstrated no consistent evidence supporting their predictive ability for long-term outcomes: high fear avoidance beliefs (from adjusted data) and female gender, the presence of previous history of LBP and pain radiating to the leg (from adjusted and unadjusted data) (Table 2 and Table 3). These variables did not reflect relevant disagreements of inverse association. On the other hand, there were 35 prognostic factors reported by a single systematic review, with insufficient evidence for synthesis (Appendix A). 

#### 3.4.2. Chronic Phase of LBP

There were four variables reported by a single systematic review and, therefore, with insufficient evidence for synthesis [35,36,40,41]: physical activity, abdominal muscle function, fear avoidance beliefs and pain catastrophism. The evidence in all of them ranged from non-association to association with the results (Appendix A). There was only one finding indicating that high fear avoidance beliefs predicted better low back pain outcomes. However, the sample size of this study was small and the follow-up short, so this may be a potentially biased finding.

#### 3.4.3. Mixed-Duration LBP

Pain catastrophism was reported by two reviews based on individuals with acute to chronic LBP [36,37], reflecting that high catastrophic thinking showed a consistent association with poor long-term outcomes and clinically relevant strengths of association (from adjusted data) (Table 4). Moreover, there were three factors in acute to chronic LBP [31,35,40] and one factor in the subacute–chronic population [34] reported by a single systematic review: physical activity, fear avoidance beliefs, work social support and recovery expectations, respectively. Once again, the evidence ranged from association to no significant association with outcomes in each one of them (Appendix A). 

On the other hand, only four factors were analyzed in the acute, subacute and chronic low back pain phases. Recovery expectations were systematically associated with outcomes regardless of the duration of symptoms, and pain catastrophism showed a trend towards association in all phases, although not always significantly. Physical activity showed a tendency of non-association in the different phases, and fear avoidance beliefs were more significant in the subacute phase of LBP.

The considerable clinical and methodological heterogeneity of the collected data (prognostic factors, outcomes and their measurements, as well as the wide range of time in the long-term follow-up) precluded the use of a meta-analysis. 

## 4. Discussion

This umbrella review provides a summary of up-to-date and high-level research evidence about biopsychosocial predictors in individuals with NSLBP. We included 15 systematic reviews, showing primary research spanning the last three decades.

A variety of biopsychosocial prognostic factors have been investigated but, in accordance with the evidence derived from the present umbrella review, only high levels of pain intensity and disability, high emotional distress, negative recovery expectations, high pain catastrophism and high work physical demands are predictors of poor low back pain outcomes at long term, and low levels of education have no prognostic ability. 

### 4.1. Factors with Consistent Evidence of Association with Poor Outcomes at Long Term

#### 4.1.1. Acute–Subacute LBP

In the present umbrella review, the factors found to be associated with poor outcomes in this phase of LBP are largely in line with the literature on LBP [15,44,45,46,47,48] and MSK [49,50,51] prognosis. In spite of this, we consider that the results suggesting that high baseline pain intensity and disability levels predict LBP outcomes should be understood from the perspective of their interactions with the factors that we discuss below. We found that individuals with high levels of emotional distress are at a greater risk of developing chronic pain and disability, with depression being the predictor with the greatest strength of association. However, its predictive capacity for the maintenance of chronic low back pain, beyond its association derived from cross-sectional studies, has been less reported in longitudinal studies, as this umbrella review shows. Nevertheless, a recent review with qualitative data on chronic LBP showed that depression had moderate evidence of no association with work-related outcomes at follow-up [52]. Moreover, recovery expectations were the most consistently reported predictor in the current umbrella review, regardless of the different outcome domains considered, as well as the phase of low back pain analyzed. Similar results have been reported in individuals with conditions other than back pain, including chronic shoulder pain [53] and major orthopedic trauma [54]. In addition, we mainly found strong association strengths with poor outcomes for high work physical demands, indicating the clinical relevance of this factor in individuals with acute–subacute LBP, in line with the previous overview of LBP prognostic factors [15]. However, two recent reviews in populations with MSK pain found insufficient evidence for physical workload [49,51] that may suggest the greater relevance of these aspects for the low back region specifically.

#### 4.1.2. Mixed Duration of LBP

Despite our results reflecting consistent evidence that high pain catastrophism predicts a delay in the functional recovery of individuals with acute to chronic LBP, as well as a trend of association with poor outcomes in the other phases of LBP, the role of catastrophic thinking remains controversial. A systematic review and meta-analysis of mediation studies suggested that “catastrophism may not explain the development of disability from back and neck pain” [55]. Moreover, it has been recently reported that pain-related acceptance is a significant mediator both between pain and catastrophism and between catastrophism and fear avoidance beliefs in chronic pain patients [56]. Thus, more studies are needed to understand the cognitive processes in the experience of pain.

### 4.2. Factors with Consistent Evidence of No Association with Poor Outcomes at Long Term in Acute–Subacute LBP

We found that a lower level of education was not associated with worse work-related outcomes, being in line with the evidence provided by previous reviews in LBP [15,48,52] and musculoskeletal populations [49]. 

### 4.3. Factors with Inconsistent Evidence of Association with Poor Outcomes at Long Term in Acute–Subacute LBP

The inconsistent evidence found for the female gender was mainly due to the findings reported by Agnello et al., but whose significant heterogeneity was explained by the compensation status of the individuals to participate in the study [30]. Considering this, our findings are consistent with the non-association evidence reported by other authors in LBP and MSK pain [15,49,51]. On the other hand, sciatica or nerve root exam results showed consistent evidence of association with poor acute–subacute LBP outcomes in a previous overview [15]. Our findings of inconsistent evidence for pain radiating to the leg could be related to the fact that the included reviews did not provide an explicit definition and their measurements ranged from LBP assessment with or without radiating pain to the assessment of neurocompressive radiculopathy. Moreover, in both the current umbrella review and the prior overview of prognostic factors in LBP [15], having previous episodes of low back pain showed inconsistent evidence of association with acute–subacute low back pain outcomes. The lack of consensus in the definition of recurrence versus new episodes of LBP [57] could explain in part the lack of consistency in these findings. Finally, the predictive role of fear avoidance beliefs (FABs) in the development and perpetuation of chronic pain has been systematically reviewed in samples of LBP [15] and musculoskeletal pain patients [44,58], with some conflicting results between them as well as with the present umbrella review. The concepts of fear and avoidance encompass a series of complex processes that interact over time, and this may suggest that they are linked. However, pain-related fear and avoidance behaviors are context-dependent and do not always co-occur [11]. Thus, an individual can both prioritize the goal of avoiding pain for protection, even without reporting fear [59], and can prioritize other valued life goals and confront the threat whilst self-reporting fear [11]. This confusing conception of fear related to pain and avoidance behaviors, evidenced in turn through the measurement instruments available so far [60], may partially explain the conflicting evidence found in this umbrella review, reflecting the complexity of these mechanisms. 

### 4.4. Other Factors with Insufficient Evidence of Association with LBP Outcomes at Long Term

In the current umbrella review, low work social support [31] and low social activity [27] were reported by one systematic review, showing predictive ability for poor outcomes in individuals with mixed-duration and acute–subacute LBP, respectively. A recent systematic review among individuals with chronic pain found that the most frequent aspect in explaining the effect of social support on the experience of pain was the stress-buffering effect [61]. More studies analyzing the mechanisms of interaction between social factors and disabling LBP are needed. In addition, older age is considered a common predictor of poor outcomes in LBP, musculoskeletal pain and sciatica [15,51,62]. We believe that age may influence the natural course of low back pain and more studies are needed to determine its predictive value in these individuals.

### 4.5. Strengths and Weaknesses 

We developed and registered a specific overview protocol in PROSPERO, minimizing reporting bias and giving transparency to the review process. Our search strategy was implemented in a sufficiently inclusive manner through relevant and grey literature databases, along with additional strategies such as manual searches and contact with authors (accounting for 14% of the reviews included), reflecting the evidence from original studies over the last 35 years and including a large number of participants (N = 257,208). 

The weaknesses of the present overview depend not only on the risk of bias and selective reporting of results by the primary studies, as reflected the publication biases shown in the findings derived from meta-analyses, but also on the quality of the included reviews, all of them being assessed to have minor limitations. Additionally, there was a modification from the initial protocol recorded in PROSPERO. For our outcome of LBP results at follow-up, we planned to synthesize the evidence for each primary outcome separately, but, due to insufficient evidence, we considered pain intensity, functional status, work participation and recovery outcomes together. Furthermore, at the level of this overview, the English and Spanish languages were considered as inclusion criteria, and therefore some reviews of interest may have been excluded. Moreover, the heterogeneity derived from the variability in adjustment models for confounders must be recognized. Our synthesis is also limited by the fact that we only included quantitative research studies; for this reason, several systematic reviews with qualitative data have been considered in our discussion. 

### 4.6. Implications for Clinicians and Policymakers

This umbrella review presents a synthesis of prognosis evidence on individuals with acute and subacute LBP in North America, Europe and Oceania in clinical and occupational settings. An enhanced understanding of the role of the psychosocial factors provides the opportunity for prevention, identifying patients at risk of chronicity and targeting treatments for modifiable factors [63,64,65]. Treatments in low back pain may consider the factors consistently reported in this umbrella review. Policymakers should include multidimensional interventions through public health systems [66]. 

### 4.7. Future Research

The factors presented in the present umbrella review, with consistent evidence of a prognostic association with LBP outcomes derived from adjusted data, can be taken into consideration for the development of low back pain causal explanatory models and for intervention trials in these patients. In view of the evidence collected, further research in the later phase of LBP and regarding social and socio-occupational factors is required. Future reviews that include a meta-analysis could gain a better estimate of prognostic effect sizes, assess and account for heterogeneity in the effects of prognostic factors and perform additional subgroup and sensitivity analyses.

Overall, we still need a better understanding of the complex dynamic relationships between biopsychosocial factors. 

## 5. Conclusions

The current umbrella review has identified consistent findings of up-to-date and high-level research evidence that support the ability of several biopsychological factors to predict LBP outcomes in the long term. Such factors are levels of pain intensity and disability, emotional distress, recovery expectations, pain catastrophism and physical demands at work. These variables deserve attention for inclusion in the development of low back pain explanatory models. More research on social and socio-occupational factors, as well as predictors, in the chronic phase of LBP is required in order to add potential prognostic information to this condition. Our findings implicate a multidimensional approach in dealing with these individuals. 

## Figures and Tables

**Figure 1 ijerph-19-10145-f001:**
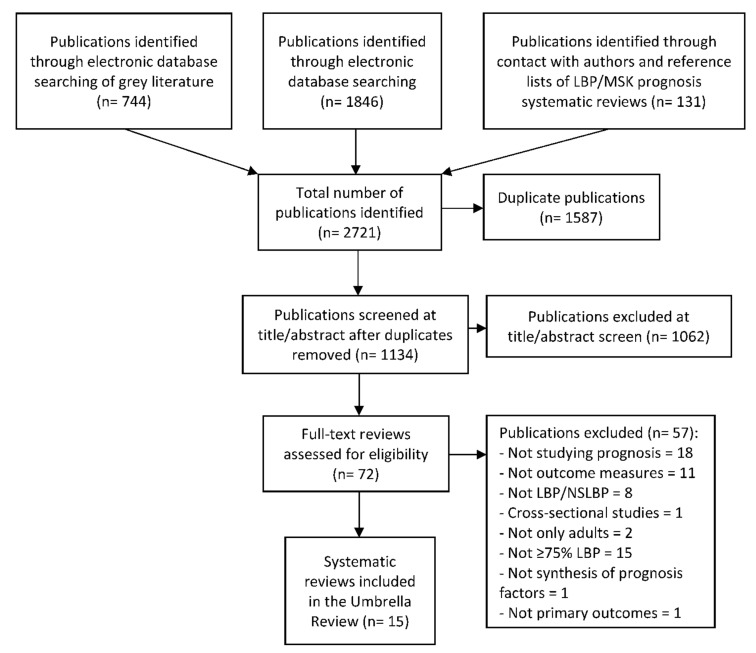
Flow chart of low back pain prognosis systematic reviews.

**Table 1 ijerph-19-10145-t001:** Characteristics of the included low back pain prognosis systematic reviews.

		Research Question					Data Extraction				
Review	Review Quality	Population/Setting	Prognostic Factor(s)	Prognosis Outcome(s)/Follow-Up: Minimum Criteria; Result	Literature Search (Citations Found)	Study Selection Criteria (Studies/Publications)/Total Participants	Associations	Prognostic Factor Categorization: Number and Type	Quality Assessment Criteria	Synthesis Strategies	Main Conclusions of the Authors
Kent PM et al., 2008 [27]	Reliable	Adults with recent-onset non-specific low back pain (<3 months), not necessarily first episode/clinical and occupational population	Biopsychosocial prognostic factors of screening instruments	Pain intensity, activity limitation and participation restriction/S/T: <3 months and L/T: >3 months; NA	MEDLINE, CINAHL, Embase, PsycINFO and AMED from inception to February 2007; reference lists of included studies and relevant reviews; citation tracking of authors of relevant studies (3881)	RQ; English; prospective cohort studies; reporting statistical association information; excluding studies with participants with specific diseases, pregnancy or more than 15% with compressive symptoms, cross-sectional, incidence/prevalence studies or describing clinical course without prognostic factor data (50–54)/33,089	1. SS+, SS−, NS 2. Effect sizes and CIs calculated; bivariate and multivariate results	1. psychosocial 2. history 3. pain 4. physical impairment 5. activity limitation 6. participation restriction 7. clinical 8. therapeutic response	List of 6 quality criteria recommended by Hudak et al., 1996 [42] with a score from 0 to 6	Count of significant results; meta-analysis	It remains uncertain which factors are associated with specific outcomes, the strength of those associations and the degree of confusion among prognostic factors.
Chou R et al., 2010 [28]	Reliable	Adults with low back pain <8 weeks/clinical (primary care, specialty or physical therapy clinics) and occupational population	Biopsychosocial factors	Chronic low back disability (pain, disability, work status, mixed results)/S/T: 3 to 6 months and L/T: ≥1 year; ranged from 3 months to 2 years	MEDLINE (1966–January 2010) and Embase (1974–February 2010); reference lists of collected studies (11,841)	RQ; English; adults; prospective cohort studies of individual risk factors or risk predictors of persistent disabling DL (14–16)/10,842 participants	1. SS+, SS−, NS 2. multivariate results mainly	1. demographic and work-related characteristics 2. health status at the beginning of the LD 3. signs and symptoms	List of 8 quality criteria	Individual study results described; meta-analysis	The most useful components for predicting persistent disabling NSLBP were lower levels of fear avoidance and low basal functional impairment, along with non-organic signs, general health status and the presence of psychiatric co-morbidities
Steenstra IA et al., 2011 [29]	Reliable	Adults with acute non-specific low back pain (<6 weeks)/clinical and occupational population	Biopsychosocial factors	Return to work/NR; varies from 2 to 264 months	MEDLINE (1966–April 2011), Embase and PsycINFO (from inception to April 2011); reference lists of relevant and recently published studies (4449)	RQ; cohort studies (prospective, retrospective) and secondary RCT analyses; results measured in absolute terms (rate), relative terms (OR, RR, HR), survival curve or duration of sick leave (25–30)/112,797 participants	1. SS+, SS−, NS 2. Effect sizes and CIs; univariate and multivariate results	International Classification of Functioning, Disability and Health (ICF) 1. factors related to the LD 2. to the worker, 3. to the work and the workplace 4. to the psychosocial environment	List of 6 quality criteria based on existing lists with a classification of high, moderate or low quality	Individual study results described; levels of evidence (strong, moderate and insufficient)	Workers’ expectations of recovery are important factors in predicting a return to work. Pain and disability factors remain important barriers to recovery. Offering modified tasks clearly helps workers return to work. However, job physical demands prevent workers from returning to work.
Agnello A et al., 2010 [30]	Reliable	Adults with acute non-specific low back pain (≤6 weeks)/clinical and occupational setting	Biopsychosocial factors	Recovery (presence or not of pain or work-related or non-work-related disability)/6 months; ranged from 6 months to 1 year	MEDLINE, CINAHL, Embase and PsycINFO from inception to November 2007; reference lists of relevant studies (2341)	RQ; English and French; subjects aged 18–65 years with radiated or non-radiated pain; occupational setting; minimum follow-up 6 months; excluding fractures and dislocations (7–10)/2484 participants	1. SS, NS 2. Effect sizes and CIs calculated; univariate results	NR	Adapted tool by Walton et al., 2009 [43] of 17 criteria, with a maximum score of 34 points and a high, moderate or low quality rating	Individual study results described; meta-analysis with adequate graphical representation	The ability of the female gender to predict the outcome is not yet clear. Pain radiating to the leg and a history of back pain have no statistical evidence to support their isolated application in clinical practice.
Campbell P et al., 2013 [31]	Reliable	Adults with mixed duration from acute to chronic non-specific low back pain/occupational setting	Work social support (general work support, co-worker and supervisor support)	Recovery results (pain intensity, disability) and return to work/NR; ranged from 6 weeks to 4 years	MEDLINE, Embase, PsycINFO, CINAHL, IBSS, AMED and BNI from inception to 18 November 2011; reference lists of recent relevant studies and reviews; citation search for validated social support measures; databases of local experts (447)	RQ; English; prospective cohort and case–control studies; excluding studies addressing informal family or social support, specific health problems, specific pregnancy or DL, cross-sectional findings and small case series (<30 persons) (13)/8091	1. SS+, SS−, NS 2. Effect sizes and CIs; univariate and multivariate results	NR	List of 16 quality criteria based on the combination of assessments of several recent reviews and guidelines for quality assessment in systematic reviews in LBP	Individual study results described; count of results based on direction effect with ranges of effect sizes	Work-related social support had a weak prognostic effect on NSLBP outcomes and may be subject to the influence of broader concepts related to the employment context.
Iles RA et al., 2009 [32]	Reliable	Adults with non-chronic non-specific low back pain (<3 months)/clinical and occupational population	Recovery expectations	Activity limitation and participation restriction (ICF)/NR; ranged from 6 weeks to 2 years	MEDLINE, Embase, PsycINFO, CINAHL, AMED, The Cochrane Library, PEDro from inception to September 2007; reference lists of included studies and relevant systematic reviews (111)	RQ; English; published in peer-reviewed journals; baseline cohorts with >75% participants with DLNS; reporting predictive strength data; excluding retrospective studies (10)/4038 participants	1. SS, NS 2. Effect sizes and CIs; univariate and multivariate results	NR	List of 14 criteria derived from 2 systematic reviews on prognosis of NSLBP, with a classification of high quality (if 10 or more criteria were satisfied) and lower quality (less than 10 criteria satisfied)	Individual study results described; count of significant results with ranges of effect sizes; graphical presentation	Expectations of recovery when measured with a specific, time-based measure within the first 3 weeks of NSLBP are a strong predictor of people at risk of poor outcomes.
Hallegraeff JM et al., 2012 [34]	Reliable	Adults with acute and subacute non-specific low back pain (<12 weeks)/occupational setting	Recovery expectations	Absence from usual work/NR; ranged from 3 to 24 months	PubMed, MEDLINE, Embase, PEDro since 1999; reference lists of studies included (591)	IP; English; prospective cohort studies and secondary RCT analyses; adults 18–65 years; living in Western (industrialized) country; OR or HR analyses; excluding studies with participants with rheumatic disease, cancer or trauma (10)/4683 participants	1. SS, NS 2. Effect sizes and Cis calculated; univariate and multivariate results	NR	List of 9 quality criteria (AHRQ), with scores below 4 indicating low risk of bias, between 4 and 6 medium risk and 7 or more high risk of bias	Individual study results described; meta-analysis with ranges of effect sizes and adequate graphical representation	Consistent evidence that negative expectations regarding early recovery are a strong predictor of future absence from usual work.
Hayden JA et al., 2019 [35]	Reliable	Adults with acute (<6 weeks), subacute or chronic (≥6 weeks) and mixed-duration non-specific low back pain/clinical and occupational setting	Recovery expectations (general, self-efficacy and treatment expectations)	Work participation, important recovery, functional limitations, pain intensity; global improvement, health-related quality of life, satisfaction with treatment, mood and healthcare use/3 months; ranged up to >16 months	MEDLINE, Embase, CINAHL, PsycINFO from inception to 12 March 2019; reference searches of relevant reviews; reference lists of included studies; citation searches of recovery expectation measurement tools; personal files of recovery expectation investigators (7235)	RQ; prospective or retrospective studies, secondary RCT analysis and associations from moderate analysis; excluding specific pathologies or conditions (60–85)/30,530 participants	1. SS+, SS−, NS 2. Effect sizes and CIs calculated; univariate and multivariate results; adjusting factors noted	NR	Quality In Prognosis Studies (QUIPS) tool with 6 domains, rated as low, moderate or high risk of bias	Individual study results described; meta-analysis with ranges of effect sizes and adequate graphical representation; GRADE quality levels of evidence	Individual recovery expectations are probably strongly associated with future work participation (moderate-quality evidence) and may be associated with clinically important recovery outcomes (low-quality evidence). The association of recovery expectations with functional limitations and pain intensity outcomes is less certain.
Wertli and Rasmussen-Barr, 2014 [36]	Reliable	Adults with acute, acute–subacute, subacute, chronic and mixed-duration non-specific low back pain/clinical and occupational setting	Fear avoidance beliefs	Work-related (days off, return to work, etc.) and non-work-related measures (pain, perceived disability, etc.)/3 months; ranged from 3 months to 2 years	BIOSIS, CINAHL, The Cochrane Library, Embase, OTSeeker, PEDro, PsycINFO, PubMed/MEDLINE, Scopus and Web of Science from 1990 to October 2011; reference lists of collected studies and manual electronic search of 6 relevant journals (2070)	RQ; no language or setting limits; using FABQ and TSK scales; cohort studies (prospective, retrospective) and secondary RCT analyses; at least moderate quality and 100 subjects; minimum follow-up 3 months; excluding conference proceedings (21)/5467 participants	1. SS, NS 2. Effect sizes and CIs; univariate and multivariate results	NR	Methodological checklist SING with a high (++), moderate (+) or low (−) quality grading	Individual study results described; count of significant results; graphical presentation	Evidence suggests that fear avoidance beliefs are predictive of poor outcome in patients with subacute NSLBP and should be addressed in this population to avoid delay in recovery.
Wertli and Eugster, 2014 [37]	Reliable	Adults with acute, acute–subacute, chronic and mixed-duration non-specific low back pain/clinical and occupational setting	Catastrophism	Work-related measures (days off, return to work, etc.) and non-work-related measures (pain, perceived disability, etc.)/3 months; ranged from 90 to 2160 days	BIOSIS, CINAHL, The Cochrane Library, Embase, OTSeeker, PEDro, PsycINFO, MEDLINE, Scopus and Web of Science from January 1980 to September 2012; reference lists of included studies, reviews and treatment guidelines; handsearching of 6 relevant journals (1528)	RQ; no language or setting limits; cohort studies (prospective, retrospective) and secondary RCT analyses; at least moderate quality; minimum 100 patients and minimum follow-up 3 months; excluding conference proceedings (16–19)/11,330 participants	1. SS, NS 2. Effect sizes and CIs; univariate and multivariate results	NR	List of SING criteria for cohort studies with a high, moderate or low quality rating	Individual study results described; count of significant results; graphical presentation	There is some evidence that catastrophism as a coping strategy can lead to a delay in recovery. The influence of catastrophism on DL patients is not fully established.
Wertli and Burgstaller, 2014 [38]	Reliable	Adults with mixed-duration from acute to chronic non-specific low back pain/clinical setting	Catastrophism	Work-related (days off, etc.) and non-work-related (pain, perceived disability, etc.) measures/NR; ranged from 7 days to 1 year	BIOSIS, CINAHL, The Cochrane Library, Embase, OTSeeker, PEDro, PsycInfo, MEDLINE, Scopus and Web of Science from January 1980 to September 2012; reference lists of included studies and handsearching of 6 relevant journals (1528)	RQ; no language or setting limits; secondary RCT analyses with a minimum of 30 patients per group; excluding conference proceedings (6–7)/1049 participants	1. SS, NS 2. Effect sizes and CIs; univariate and multivariate results, if available	NR	List of SING criteria for RCTs with a high, moderate or low quality rating	Individual study results described; count of significant results	Catastrophism predicted outcomes for pain and disability at follow-up in patients with NSLBP.
Pinheiro MB et al., 2016 [39]	Reliable	Adults with acute or subacute non-specific low back pain (<3 months)/clinical and occupational setting	Depression	Work-related measures, pain intensity, disability, self-perceived recovery and mixed/unrestricted; ranged from 2 weeks to >12 months	AMED, CINAHL, Embase, Health & Society Database, LILACS, MEDLINE, PsycINFO, Scopus and Web of Science from inception to 10 October 2014; reference lists of included studies and systematic reviews (10,541)	RQ; no limits on language, setting, length of follow-up or type of publication; prospective cohort studies; excluding pregnancy-specific or pregnancy-related LBs and secondary analyses of RCTs (13–17)/5396 participants	1. SS+, SS−, NS 2. Effect sizes with CIs; univariate and multivariate analysis; adjusting factors noted and detailed	NR	List of 8 criteria based on recommendations for systematic reviews and the STROBE guide	Individual study results described; count of significant results; graphical presentation	Depression might have an adverse effect on the prognosis of low back pain.
Hendrick P et al., 2011 [40]	Reliable	Adults with mixed-duration from acute to chronic non-specific low back pain/NR	Physical activity in daily life (occupational, sports and leisure activities)	Pain, disability and number of health treatments results in 1 year/NR; ranged from 1 to 5 years	OVID, CINAHL, MEDLINE, AMED, Embase, Biomed, PubMed—National Library of Medicine, Proquest and The Cochrane Library from 1990 to January 2009; reference lists of included studies; experts and authors of included studies contacted (405)	RQ; English; >18 years; cohort studies, secondary RCT and case–control analyses; excluded retrospectives (7)/3535 participants	1. SS, NS 2. Effect sizes and CIs; multivariate results mainly	NR	Modified Down and Black list of 23 items, with a maximum score of 27 points	Individual study results described; count of significant results	The results provide moderate evidence that activity or change in activity in patients with NSLBP is not predictive of LBP outcomes.
Oliveira CB et al., 2019 [41]	Reliable	Adults with acute, subacute and chronic non-specific low back pain/clinical and general population	Physical activity (any type)	Results for pain intensity, disability and recovery measures/unrestricted; ranged from 3 months to 5 years	MEDLINE, Embase, CINAHL, SPORTDiscus and Web of Science from inception to February 2018; reference lists of included studies and systematic reviews (12,681)	RQ; English, Spanish, Portuguese; prospective cohort studies; excluding secondary RCT analyses (12)/8455 participants	1. SS+, SS−, NS 2. Effect sizes and CIs; univariate and multivariate analysis; adjusting factors noted	NR	Quality In Prognosis Studies (QUIPS) tool with 6 domains, rated as low, moderate or high risk of bias	Individual study results described; GRADE quality levels of evidence (high, moderate, low and very low)	There was low-quality evidence that physical activity may not be a factor in predicting pain, disability or recovery outcomes in NSLBP.
Wong AY et al., 2013 [33]	Reliable	Adults with acute, subacute and chronic non-specific low back pain/clinical (hospital, general practice clinics and physical therapy) and general populations	Characteristics of TrA and LM assessed by dynamic morphometry, histology and muscle activation	Pain and function results/NR; ranged from 1 week to 1 year	MEDLINE, Embase, PEDro, SPORTDiscus, CINAHL and The Cochrane Library from inception to December 2012; ClinicalTrials.gov, NIH Clinical Center Clinical Research Studies and Current Controlled Trials Register; contact with investigators or authors (2325)	RQ; English, Chinese, French, Portuguese; cohort studies (prospective, retrospective), secondary RCT analyses, case series with 10 or more subjects, systematic reviews or meta-analyses (5)/219 participants	1. SS, NS 2. Effect sizes and CIs; multivariate analysis only; adjusting factors noted	NR	Adapted criteria list with 7 potential bias areas with a maximum score of 26 points and a cut-off point of 50% of the total score indicating high quality	Individual study results described; levels of evidence (strong, moderate, limited, conflicting and non-evidence)	There was conflicting evidence regarding the dynamic morphometry of TrA/LM when predicting low-back-pain-related disability or pain reduction in patients with chronic non-specific low back pain after various conservative treatments.

NSLBP = non-specific low back pain; NR = not reported; RQ = review search question; SS = statistically significant; NS = not statistically significant; CI = confidence interval; S/T = short term; L/T = long term; TrA: transversus abdominis; LM: lumbar multifidus.

**Table 2 ijerph-19-10145-t002:** Results of prognostic factors for LBP outcomes at long term, reported by two or more systematic reviews using OR/beta coefficients.

Prognostic Factor Domain	Prognostic Factor	Factor Definition	Author, Year [Ref]	Nº Primary Studies Included (N) ^Ref.^	Outcome	Adjusted OR/Beta	Crude OR/Beta	Heterogeneity Q Statistic (*p*)	Publication Bias
**Acute and Subacute LBP (≤3 months)**									
**Factors related to the individual**	Gender	Gender (Female)	Agnello A, 2010 [30] *	6 studies (N = 2306) ^1–6^	Ra		Pooled OR = **1.28, 95% CI = 1.03–1.58** (*p* = 0.02)	Q = 14.6 (*p* = 0.01)	The failsafe N = 4 (ss)
			Kent PM, 2008 [27]	2 studies (N = 334) ^7,8^	P	Pooled OR = 1.97, 95% CI = 0.98–3.97 ***		NR	NR
				3 studies (N = 833) ^8–10^	FS	Pooled OR = 1.38, 95% CI = 0.64–2.99		NR	NR
				2 studies (N = 1154) ^6,11^	WP	Pooled OR = 0.61, 95% CI = 0.30–1.24		NR	NR
	Education	Lower education level	Steenstra IA, 2011 [29] *	2 studies (N = 2739) ^11,12^	WP	OR = 0.92, 95% CI = 0.55–1.54		NA	NA
		Lower education level	Kent PM, 2008 [27]	2 studies (N = 1.114) ^11,13^	WP	Pooled OR = 0.99, 95% CI = 0.63–1.55		NR	NR
**Factors related to the episode**	Prior episodes	Previous history of low back pain (yes/no)	Agnello A, 2010 [30] *	3 studies (N = 382) ^3,4,6^	Ra		Pooled OR = 0.91, 95% CI = 0.52–1.60 (*p* = 0.75)	Q = 1.64 (*p* = 0.44)	The failsafe N (ns)
			Kent PM, 2008 [27]	2 studies (N = 818) ^10,14^	FS	Pooled OR = **2.98, 95% CI = 1.42–6.23**		NR	NR
				2 studies (N = 1154) ^6,11^	WP	Pooled OR = 0.99, 95% CI = 0.39–2.53		NR	NR
**Pain**	Pain radiating to the leg	Pain radiating to the leg (yes/no)	Agnello A, 2010 [30] *	4 studies (N = 502) ^3,4,6,15^	Ra		Pooled OR = 1.37, 95% CI = 0.79–2.39 (*p* = 0.26)	Q = 5.99 (*p* = 0.11)	The failsafe N (ns)
		Pain radiating to leg (yes/no)	Steenstra IA, 2011 [29] *	3 studies (N = 1421) ^16–18^	WP	OR ranged from **4.9, 95% CI = 2.8–8.4** to **6.25, 95% CI = 4.42–8.96**	OR = **2.5, 95% CI = 1.1–5.8**	NA	NA
		Severity of leg pain (ref. mild sprain/strain: major sprain/strain—radiculopathy)		1 study (N = 1885) ^12^	WP	OR = **3.72, 95% CI = 1.83–7.58**		NA	NA
		Intensity of leg pain (7–10)		1 study (N = 854) ^11^	WP		OR = **1.92, 95% CI = 1.11–3.33**	NA	NA
		Leg pain (yes/no)	Kent PM, 2008 [27]	1 study (N = 219) ^7^	P	Largest significant OR = **2.45, 95% CI = 1.20–4.99**		NA	NA
				3 studies (N = 938) ^9,10,15^	FS	Largest significant OR = **3.30, 95% CI = 1.10–9.60**		NA	NA
				2 studies (N = 1154) ^6,11^	WP	Pooled OR = 2.10, 95% CI = 0.96–4.62		NR	NR
	Pain intensity	Greater pain intensity	Kent PM, 2008 [27]	1 study (N = 542) ^9^	FS	Largest significant OR = **2.84, 95% CI = 1.70–4.80**		NA	NA
				3 studies (N = 1334) ^6,11,19^	WP	Pooled OR = **1.45, 95% CI = 1.10–1.91**		NR	NR
		Pain interference with daily activities	Steenstra IA, 2011 [29] *	2 studies (N = 532) ^16,20^	WP	OR ranged from **1.57, 95% CI = 1.27–1.94** to **4.7, 95% CI = 1.8–12.5**		NA	NA
		Number of sites with pain (0–2/3–4/≥5)		1 study (N = 1885) ^12^	WP	OR = **1.71, 95% CI = 1.01–2.92**		NA	NA
		Pain change (better/unchanged/worse)		1 study (N = 1885) ^12^	WP	OR = 1.47, 95% CI = 0.98–2.20		NA	NA
		Greater pain intensity (mild/moderate/severe)		1 study (N = 854) ^11^	WP		OR = 1.47, 95% CI = 0.74–2.91	NA	NA
**Functional limitation**	Disability	High self-reported disability (RMDQ, ODI, others)	Steenstra IA, 2011 [29] *	4 studies (N = 3247) ^11,12,20,21^	WP	OR ranged from **1.40, 95% CI = 1.05–6.63 to 7.01, 95% CI = 3.44–14.29**	Unclear	NA	NA
		High score on Oswestry Disability Index	Kent PM, 2008 [27]	1 study (N = 130) ^19^	FS	Largest significant OR = **3.35, 95% CI = 1.42–2.37**		NA	NA
				3 studies (N = 1334) ^6,11,19^	WP	Pooled OR = **2.69, 95% CI = 1.01–7.15**		NR	NR
**Psychological–emotional**	Emotional distress	Depression (high scores)	Kent PM, 2008 [27]	1 study (N = 138) ^22^	P	Largest significant OR = **28.70, 95% CI = 3.52–233.91**		NA	NA
				2 studies (N = 1154) ^6,11^	WP	Pooled OR = **1.50, 95% CI = 0.48–4.71**		NR	NR
		Symptoms of depression (presence/higher scores)	Pinheiro MB, 2016 [39]	1 study (N = 315) ^23^	P	OR = **1.03, 95% CI = 0.98–1.08**		NA	NA
				2 studies (N = 573) ^24,25^	FS	OR = **1.06, 95% CI = 1.02–1.11 and β = 0.20, 95% CI = 0.04–0.36**		NA	NA
				4 studies (N = 1909) ^11,26–28^	WP	OR = **1.10, 95% CI = 1.04–1.17**	Unclear	NA	NA
				1 study (N = 439) ^29^	Rb	OR = **1.06, 95% CI = 1.03–1.09**		NA	NA
		Anxiety (high scores)	Kent PM, 2008 [27]	2 studies (N = 2712) ^7,30^	P	Largest significant OR **= 2.68, 95% CI = 1.28–5.58**		NA	NA
				1 study (N = 854) ^11^	WP	Largest significant OR **= 2.08, 95% CI = 1.50–2.89**		NA	NA
**Psychological–cognitive**	Fear avoidance beliefs	High fear avoidance beliefs (FABQ)	Steenstra IA, 2011 [29] *	2 studies (N = 2953) ^5,12^	WP	Unclear		NA	NA
		High fear avoidance beliefs (FABQ and TSK)	Wertli MM, 2014 [36] *	3 studies (N = 637) ^4,31,32^	Ra	Unclear		NA	NA
				1 study (N = 940) ^12^	WP	OR = 1.71, 95% CI = 0.88–3.3		NA	NA
		High fear avoidance beliefs—Physical activity (FABQ-P)	Wertli MM, 2014 [36]	1 study (N = 171) ^33^	FS	OR = 1.73, 95% CI = 0.6–4.99		NA	NA
				1 study (N = 171) ^33^	Ra	OR = 1.58, 95% CI = 0.7–3.53		NA	NA
		High fear avoidance beliefs—Work (FABQ-W)		2 studies (N = 1507) ^5,34^	WP	OR ranged from **3.13 (NR), *p* = 0.00** to **4.64, 95% CI = 1.57–13.71**		NA	NA
		High fear avoidance beliefs (FABQ)	Kent PM, 2008 [27]	1 study (N = 300) ^6^	WP	Largest significant OR **= 2.77, 95% CI = 1.02–7.55**		NA	NA
		Low fear avoidance beliefs (FABQ-W)	Wertli MM, 2014 [36] **	1 study (N = 258) ^35^	WP	OR = **0.38, 95% CI = 0.25–0.58**		NA	NA
		High fear avoidance beliefs (FABQ)		1 study (N = 346) ^20^		OR = **1.05, 95% CI = 1.02–1.09**			NA
	Recovery expectations	Low recovery expectations (how likely it is that they will return to work/how long it will be before they are able to return)	Steenstra IA, 2011 [29] *	5 studies (N = 2326) ^5,16,20,36–38^	WP	OR ranged from **1.14, 95% CI = 1.04–1.25 to 3.8, 95% CI = 1.46–6.48**	OR ranged from **1.22, 95% CI = 1.02–1.45** to **4.6, 95% CI = 2.1–10.3**	NA	NA
		Negative recovery expectations (general expectations of recovery and self-efficacy)	Iles RA, 2009 [32]	1 study (N = 156) ^39^	FS	OR = **2.21, 95% CI = 1.54–2.89**		NA	NA
				7 studies (N = 2321) ^5,36,37,40–42^	WP	OR ranged from **1.21, 95% CI = 1.01–1.45 to 3.9, 95% CI = 1.77–5.38**	OR ranged from **2.3, 95% CI = 1.4–3.8 *p* = 0.001** to **9.18, 95% CI = 5.00–16.8**	NA	NA
		Negative recovery expectations (general expectations of recovery and self-efficacy)	Hallegraeff JM, 2012 [34]	10 studies (N = 4649) ^5,36–38,40–45^	WP	Pooled OR = **2.17, 95% CI = 1.61–2.91 *****		Q = 96.23 (*p* < 0.0001)	NA
**Occupational**	Work physical demands	High work physical demands (occupation)	Steenstra IA, 2011 [29] *	2 studies (N = 3605) ^20,46^	WP	OR ranged from **1.88, 95% CI = 1.12–3.17** to **2.27, 95% CI = 1.21–3.92**		NA	NA
		High work physical demands—self-reported (lift, bend, twist)		2 studies (N = 1016) ^11,21^	WP	OR = **3.23, 95% CI = 1.50–6.97**	OR = **1.98, 95% CI = 1.30–3.04**	NA	NA
		Job physically demanding	Kent PM, 2008 [27]	1 study (N = 120) ^15^	FS	Largest significant OR= **4.00, 95% CI = 1.10–14.00**		NA	NA
				1 study (N = 854) ^11^	WP	Largest significant OR= **2.04, 95% CI = 1.41–2.96**		NA	NA
**Acute to chronic LBP**									
**Psychological–cognitive**	Pain catastrophism	Pain catastrophism (High) (CSQ, PRSS, PCC)	Wertli MM, 2014 [38]	2 studies (N = 474) ^47–49^	FS	Standardized β ranged from **0.25; 95% CI 0.12–0.35 to 0.43, 95% CI = 0.25–0.61**	NA	OR = **1.77, 95% CI = 1.13–2.75**	NA
			Wertli MM, 2014 [37]	3 studies (N = 3423) ^48,50,51^	FS	OR ranged from **0.64, 95% CI = 0.4–0.96** (for decrease RMQ ≥ 30%) to **7.63, 95% CI = 3.69–15.7**	NA		

LBP = low back pain; N = sample size; ^ref.^ = references provided in Appendix A; OR = odds ratio; NR = not reported; NA = not assessable; ss = significant result; ns = non-significant result. Outcome: P = pain; FS = functional status; WP = work participation; R (a, b, c, d): recovery a = recovery of pain or disability, b = self-reported recovery, c = slightly better” or “worse” score on two or more follow-up measurements, d = recovery and/or return to work. * Sample of individuals in acute phase of low back pain. ** Sample of individuals in subacute phase of low back pain. *** Meta-analysis combining adjusted and adjusted data. Bold results are statistically significant.

**Table 3 ijerph-19-10145-t003:** Prognostic factors for LBP outcomes reported by two or more systematic reviews at long term.

Prognostic Factor	Kent PM, 2008 [27]	Steenstra IA, 2011 [29]	Agnello A, 2010 [30]	Pinheiro MB, 2016 [39]	Wertli MM, 2014 [36]	Wertli MM, 2014 [37]	Wertli MM, 2014 [38]	Iles RA, 2009 [32]	Hallegraeff JM, 2012 [34]	Total	Associated with Poor Outcome (+)	Not Associated with Outcome (Ø)	Unclear Evidence	Consistent Conclusions
** *ACUTE AND SUBACUTE LBP* **														
**Adjusted data**														
Level of education	Ø	Ø									0	2	0	✓
Pain intensity	+	+									2	0	0	✓
Disability	+	+									2	0	0	✓
Emotional distress	+			+							2	0	0	✓
Fear avoidance beliefs	+	Unclear			Unclear						1	0	2	
Recovery expectations		+						+	+		3	0	0	✓
Work physical demands	+	+									2	0	0	✓
**Adjusted and unadjusted data**														
Gender	Ø		+								1	1	0	
Previous history of LBP	Unclear		Ø								0	1	1	
Pain radiating to the leg	Unclear	+	Ø								1	1	1	
** *ACUTE TO CHRONIC LBP* **														
**Adjusted data**														
Pain catastrophism						+	+				2	0	0	✓

LBP = low back pain. “+”: prognostic factor with consistent association with LBP outcome; “Ø”: factors not associated with outcome; unclear: conflicting or insufficient evidence; ✓: Factor consistently associated with LBP outcomes.

**Table 4 ijerph-19-10145-t004:** Predictor variables of an explanatory model in patients with LBP.

Prognostic Factor Domain	Prognostic Factor
** *ACUTE AND SUBACUTE LBP* **	
**Pain**	Pain intensity
**Functional limitation**	Disability
**Psychological**	Emotional distress
	Recovery expectations
**Occupational**	Work physical demands
** *ACUTE TO CHRONIC LBP* **	
**Psychological**	Pain catastrophism

LBP = low back pain.

## Data Availability

No new data were created or analyzed in this study. Data sharing is not applicable to this article.

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
