# Peer review of "Biopsychosocial Factors for Chronicity in Individuals with Non-Specific Low Back Pain: An Umbrella Review"

_ijerph, 2022, doi:10.3390/ijerph191610145_

Round 1

Reviewer 1 Report

The authors did a great job with a very important and scientifically important theme.

I have just the following correction:

Table 2: there are some parts written in Spanish (agudo y subagudo; meses)

Author Response

Response to Reviewer 1 Comments

Comments and Suggestions for Authors

The authors did a great job with a very important and scientifically important theme.

Response: Thank you for these comments.

I have just the following correction:

Table 2: there are some parts written in Spanish (agudo y subagudo; meses)

Response: Thank you for pointing this out. We have followed the reviewer’s indications and edited Table 2 as follows:

Acute and Subacute LBP (≤3 months)

Reviewer 2 Report

the objective of this umbrella review is to display an up-to-date overview of high-level research evidence providing longitudinal data on biopsychosocial prognos-66 tic factors of outcomes in individuals with non-specific low back pain.

L76:add PRISMA checklist as supplemental table.

L82: why did you restrict the period as 2008? Add the reason.

L102. Did you exclude “overlapping studies in the included reviews “ in the synthesis?

L109: “acute-subacute (≤ 3 months), chronic (> 3 months)”, add the citation.
L120. Did you consider crude OR
adjusted OR as more important value?
L148: “odd ratio” means unadjusted OR?

L156: did you do subgroup or sensitivity analysis?

Author Response

Response to Reviewer 2 Comments

Comments and Suggestions for Authors

The objective of this umbrella review is to display an up-to-date overview of high-level research evidence providing longitudinal data on biopsychosocial prognos-66 tic factors of outcomes in individuals with non-specific low back pain.

L76:add PRISMA checklist as supplemental table.

Response: Thank you for pointing this out.

We have now included the PRISMA checklist as Supplemental Table S1 and the Protocol and Registration subsection has been edited as follows:

We followed the Umbrella Review Methodology Working Group [16] and considered the Preferred Reporting Items for Systematic Reviews and Meta-Analyses (PRISMA) recommendations [17] (Supplementary Table S1: PRISMA checklist). The protocol of the study was registered on PROSPERO 2020: CRD42020155081.

Additionally, we have corrected the numbering of the remaining Supplementary Tables throughout the Results subsection and Supplementary Materials, as shown below:

The search strategy included low back pain (Cochrane Back and Neck Group recommended strategy) [18] and prognostic study methods terms [19] (Supplementary Table S2: Search strategy for PubMed).

References from excluded full-text citations (n=57) are reported in the Supplementary Table S3. The conflicts of interests of the review authors are displayed in the Supplementary Table S4.

The publication date of the included reviews ranged from 2008 to 2019 and that of the included primary studies varied from 1981 to 2017 (Supplementary Table S5: Primary studies referenced in tables). Twelve reviews (80%) were published over 5 years ago (before 2015). The sample size in the reviews ranged from 219 [40] to 112.797 [28], with a mean of 17.147 (interquartile range (IQR): 3.535 to 11.330).

The common estimates used to report the results across the reviews were odds ratios (OR) but beta coefficients (β), risk ratios (RR), prevalence ratios (RP), hazard ratios (HR), likelihood ratios (LR+/LR-) and p-values were also reported. Data from RR/RP, HR, LR+/LR- and p-values are provided in Supplementary Table S6.

The results of our appraisal of the methodological quality (reliability) of the included reviews are shown in the Supplementary Table S7. We judged all 15 included reviews

On the other hand, there were 35 prognostic factors reported by a single systematic review, with insufficient evidence for synthesis (Supplementary Table S8).

The evidence in all of them ranged from non-association to association with the results (Supplementary Table S9).

Once again, the evidence ranged from association to no significant association with outcomes in each one of them (Supplementary Table S10).

Supplementary Materials: The following supporting information can be downloaded at: www.mdpi.com/xxx/s1, Table S1: PRISMA checklist; Table S2: Search strategy for PubMed; Table S3: Reference from excluded full-text citations; Table S4: Conflicts of interest of included studies; Table S5: Primary studies referenced in tables; Table S6: Other statistics (RR, RP, HR, LR+LR- and p values); Table S7: Reliability of included reviews; Table S8: Prognostic factors reported by only one review; Table S9: Prognostic factors in Chronic LBP; Table S10: Prognostic factors in Mixed-duration LBP.

L82: why did you restrict the period as 2008? Add the reason.

Response: We agree that these aspects should be detailed in the text.

We have included the following text within the Literature Search subsection:

PubMed, Cochrane Database of Systematic Reviews, Web of Science, PsycINFO, CINAHL Plus and PEDro were searched electronically for studies published between 1 January 2008 to 20 March 2020. Our search was limited from 2008, given that the previous "review of reviews" on LBP prognosis conducted the literature search until 2007 [15].

Additionally, we included the reference:

  1. Hayden, J.A.; Chou, R.; Hogg-Johnson, S.; Bombardier, C. Systematic Reviews of Low Back Pain Prognosis Had Variable Methods, and Results-Guidance for Future Prognosis Reviews. J. Clin. Epidemiol. 2009, 62, 781–796, doi:10.1016/j.jclinepi.2008.09.004

L102. Did you exclude “overlapping studies in the included reviews “ in the synthesis?

Response: We have considered this comment, and have now included the following text within the Data Extraction and Management subsection:

In addition, the overlap of primary studies among the included reviews was recorder using citation matrices and excluded from our synthesis. The degree of overlapping studies was calculated using the Corrected Covered Area (CCA) method [22].

L109: “acute-subacute (≤ 3 months), chronic (> 3 months)”, add the citation.

Response: We have followed the reviewer’s indications.

We have included the following reference in the Data Extraction and Management subsection:

  1. Pengel, L.H.; Herbert, R.D.; Maher, C.G.; Refshauge, K.M. Acute Low Back Pain: Systematic Review of Its Prognosis. Br. Med. J. 2003, 327, 323–325, doi:10.1136/bmj.327.7410.323

The text now reads as follows:

We recorded complete information about citation, population, method, prognostic factors, and outcomes assessed. The results of the reviews were removed separately for each duration of LBP symptoms: acute-subacute (≤ 3 months), chronic (> 3 months) and mixed-duration [21].

L120. Did you consider crude OR adjusted OR as more important value?

Response: Thank you for pointing this out.

We have realized that the expression is not appropriate in the text. It was intended to indicate that both findings (crude and adjusted) were “extracted” separately for analysis in the same way, when possible, on the basis that crude data do not control for confounding factors (unlike adjusted data).

Data Extraction and Management subsection has been edited as follows:

Moreover, since an unadjusted finding does not control for confounding factors, unlike the adjusted finding, we extracted all adjusted data apart from the unadjusted data for a separately planned analysis, when possible [41].

Additionally, we included the reference:

  1. Voils, C.I.; Crandell, J.L.; Chang, Y.; Leeman, J.; Sandelowski, M. Combining Adjusted and Unadjusted Findings in Mixed Research Synthesis. J. Eval. Clin. Pract. 2011, 17, 429–434, doi:10.1111/j.1365-2753.2010.01444.x

However, to perform our qualitative synthesis of the available quantitative findings for the factor “pain catastrophism”, in which there were insufficient data for a separate synthesis, we followed the criteria of Voils et al., 2011 according to which it is possible to proceed to the combination of crude and adjusted data to build a richer evidence base. This aspect has already been stated in detail in the Synthesis of Results subsection.

L148: “odd ratio” means unadjusted OR?

Response: We agree with reviewer’s indications, that they are due to a previous error, which has been corrected indicating that the data was extracted instead of deleted.

Since this aspect has been corrected in the Data Extraction and Management subsection, from our point of view the rest of the text is already properly understood, so we appreciate your comment.

L156: did you do subgroup or sensitivity analysis?

Response: Thank you for pointing this out. To clarify this aspect for readers, we have included the following text in the Data Synthesis subsection:

Thus, a qualitative synthesis was performed given that the main purpose of this study has been to present a summary of the current body of evidence based on systematic reviews of biopsychosocial prognostic factors in patients with LBP and also considering the heterogeneity of the data collected. In this way, we have described and discussed the extent of the main differences found in the results reported by the included reviews, as well as the aspects considered as probable explanatory factors for such heterogeneity, without performing additional subgroup or sensitivity analyses.

Furthermore, we consider that this could be a purpose for future studies, having been proposed as a line of research in the Future Research section, as shown below:

Future reviews that include a meta-analysis could gain a better estimate of prognostic effect sizes, assess and account for heterogeneity in the effects of prognostic factors, and perform additional subgroup and sensitivity analyses.

Reviewer 3 Report

The author conducted a comprehensive analysis of Biopsychosocial variables that contribute to chronicity in people with non-specific low back pain. According to their findings, there are five potential prognostic markers that might help inform explanatory models for low back pain. I believe that the current format of research work is well organized and suitably for publication in the international journal of Environmental Research and Public Health of MDPI. Therefore, I recommend that the article could be accepted for this journal with minor revisions. First, however, here are my general and specific comments to the authors.

General comments: using an umbrella review the author found that there were five prognostic factors which could be considered for inclusion in low back pain explanatory models. According to my opinion, the current study work style is well-planned and ready for publishing in MDPI's international journal of Environmental Research and Public Health. As a result, I believe the work might be accepted for publication in this Journal with just minor revisions.

 Specific comments:

1. Introduction: "Low back pain (LBP) is a common but a major health condition." This is confusing. Please rewrite the sentence.

 2. In line 249, Please correct "there were 4 variables" ".

3. The reference section should be formatted according to the Journal's instructions.

4. The level of English language and style needs minor editing of the manuscript.

Author Response

Response to Reviewer 3 Comments

Comments and Suggestions for Authors

The author conducted a comprehensive analysis of Biopsychosocial variables that contribute to chronicity in people with non-specific low back pain. According to their findings, there are five potential prognostic markers that might help inform explanatory models for low back pain. I believe that the current format of research work is well organized and suitably for publication in the international journal of Environmental Research and Public Health of MDPI. Therefore, I recommend that the article could be accepted for this journal with minor revisions. First, however, here are my general and specific comments to the authors.

Response: Thank you for your support.

General comments: using an umbrella review the author found that there were five prognostic factors which could be considered for inclusion in low back pain explanatory models. According to my opinion, the current study work style is well-planned and ready for publishing in MDPI's international journal of Environmental Research and Public Health. As a result, I believe the work might be accepted for publication in this Journal with just minor revisions.

Response: Thank you for these comments.

Specific comments:

  1. Introduction: "Low back pain (LBP) is a common but a major health condition." This is confusing. Please rewrite the sentence.

Response: We agree with the reviewer, this is confusing,

We have replaced this sentence with the one described below in the Introduction section:

Low back pain (LBP) is a common health condition with important implications for individuals, public health systems and economies [9].

Additionally, we included the reference:

  1. Hartvigsen, J.; Hancock, M.J.; Kongsted, A.; Louw, Q.; Ferreira, M.L.; Genevay, S.; Hoy, D.; Karppinen, J.; Pransky, G.; Sieper, J.; et al. What Low Back Pain Is and Why We Need to Pay Attention. Lancet 2018, 391, 2356–2367, doi:10.1016/S0140-6736(18)30480-X

  1. In line 249, Please correct "there were 4 variables" ".

Response: We have followed the reviewer's indication. In accordance with the basic rule in English that numbers from zero to nine must be expressed in words, we have proceeded to correct this aspect in the Results section.

The text now reads as follows:

There were  four variables reported by a single systematic review and therefore, with insufficient evidence for synthesis [34,35,39,40]: physical activity, abdominal muscle function, fear avoidance beliefs and pain catastrophism.

  1. The reference section should be formatted according to the Journal's instructions.

Response: Thank you for pointing this out. We have revised the References section and corrected the style in several bibliographic citations.

Additionally, we have replaced the reference of the open access institutional repository of the Methodology for JBI Umbrella Reviews with that of its subsequent publication in a journal article, as this is the latest publication available.

The reference now reads as follows:

  1. Aromataris, E.; Fernandez, R.; Godfrey, C.M.; Holly, C.; Khalil, H.; Tungpunkom, P. Summarizing systematic reviews: Methodological development, conduct and reporting of an umbrella review approach. Int. J. Evid. Based Healthc. 2015, 13, 132–140, doi: 10.1097/XEB.0000000000000055

  1. The level of English language and style needs minor editing of the manuscript.

Response: We have made edits as suggested by the reviewer. The manuscript was revised in its entirety by a native English speaker who is an expert in the area and has extensive experience in proofreading and editing texts for publication in this field.

Corrections can be viewed through the "Track Changes" function throughout the entire manuscript.
